# Design and Implementation of a Hardware and Software System for Visual Assessment of Bituminous Coating Quality

**DOI:** 10.3390/s23239325

**Published:** 2023-11-22

**Authors:** Dmitrii Kamianskii, Alexander Boldyrev, Nikita Vezdenetsky, Irina Vatamaniuk, Marina Bolsunovskaya

**Affiliations:** 1Laboratory “Industrial Systems for Streaming Data Processing”, School of Advanced Engineering Studies in Digital Engineering, Peter the Great St. Petersburg Polytechnic University, Polytechnicheskaya, 29, 195251 Saint Petersburg, Russiamarina.bolsunovskaia@spbpu.com (M.B.); 2LLC Amdor, 192148 Saint Petersburg, Russia; mail@amdor.ru

**Keywords:** asphalt mixtures, adhesion, boiling water test, image analysis, surface segmentation, gradient boosting, U-Net

## Abstract

Typically, the quality of the bitumen adhesion in asphalt mixtures is assessed manually by a group of experts who assign subjective ratings to the thickness of the residual bitumen coating on the gravel samples. To automate this process, we propose a hardware and software system for visual assessment of bituminous coating quality, which provides the results both in the form of a discrete estimate compatible with the expert one, and in a more general percentage for a set of samples. The developed methodology ensures static conditions of image capturing, insensitive to external circumstances. This is achieved by using a hardware construction designed to provide capturing the samples at eight different illumination angles. As a result, a generalized image is obtained, in which the effect of highlights and shadows is eliminated. After preprocessing, each gravel sample independently undergoes surface semantic segmentation procedure. Two most relevant approaches of semantic image segmentation were considered: gradient boosting and U-Net architecture. These approaches were compared by both stone surface segmentation accuracy, where they showed the same 77% result and the effectiveness in determining a discrete estimate. Gradient boosting showed an accuracy 2% higher than the U-Net for it and was thereby chosen as the main model when developing the prototype. According to the test results, the evaluation of the algorithm in 75% of cases completely coincided with the expert one, and it had a slight deviation from it in another 22% of cases. The developed solution allows for standardizing the data obtained and contributes to the creation of an interlaboratory digital research database.

## 1. Introduction

Throughout the world, paved roads are mainly constructed with asphalt or asphalt concrete consisting of a mixture of stones and bitumen, which is an oil substance that serves as a gluing component for stones. In order to ensure the surface robustness and prevent the stones from detachment, it is crucial to establish a stable bond between the stones and bitumen. In the case of pavement delamination, repair or repaving may entail substantial costs. Hence, prior to approval of the bituminous mixture use, preliminary laboratory tests are executed to determine its resistance to physical impact on a particular type of gravel base.

For this purpose, two test methods, namely, the rolling bottle and boiling water tests, are widely implemented [1]. The resistance of the bituminous coating to contact interaction is tested by the rolling bottle method, whereas its resistance to temperature exposure and water erosion is studied by means of boiling. After the experiments, the test samples are subjected to an assessment of the residual bitumen adhesion.

Currently, the main assessment method involves visual analysis conducted by a group of experts who independently determine discrete quality estimates for each stone in the test sample based on visual analysis of the residual coating level. The resulting quality assessment of the coating and the bituminous mixture is the product of averaging the estimates, initially by sample and then by a group of experts.

The rolling bottle method provides a simpler visual analysis of residual coating level in comparison with the boiling water method [2]. This is due to the fact that it only requires consideration of the bare gravel area resulting from chipping of the bituminous coating. When examining the stone surface after boiling, it should be noted that the results may not be entirely conclusive, as the bitumen tends to detach unevenly. Its final thickness in different areas can vary greatly, thereby affecting the assessment. In this regard, the following four surface types are identified, which categorize certain ranges of bitumen layer thickness based on their visual attributes and proportionate areas on the stone surface (see Figure 1):Bare gravel, which is the surface that lacks or has very little bituminous coating;Insufficient layer, which is the surface with a thin bitumen layer that allows the gravel underneath to be visible or has very few patches of the coating;Thin layer, which is the surface with the gravel base obscured by the bitumen layer but with clearly visible relief;Thick layer, which is the surface with the gravel base relief either completely invisible or extremely smoothed.

However, the expert assessment method has a number of significant disadvantages:

Subjective assessments: There are no strict criteria for assigning a stone coating to one or another assessment class; the assessment from one expert for the same stone may vary at different times;

Slow assessments: To compensate for the subjective factor, an expert group is involved for independent assessment, which greatly slows down the rate of obtaining results;

Insufficient assessments: During the expert assessment of a stone, only four core quality estimates are distinguished: 2, 3, 4, and 5 with the possibility of assigning an intermediate estimate, thus the stones are distributed by residual coating in seven classes. Due to the subjectivity of the assessment, the distribution into a larger number of classes appears to be extremely difficult. This makes it impossible to more accurately select the emulsion mixture ingredients in the absence of Evident visible differences among various samples.

The process automation removes the subjective factor from the assessment procedure, provides a percentage integral estimate, and significantly accelerates obtaining a result. Moreover, it allows standardization of studies across different laboratories. Our study strives to devise a hardware and software system for automated quality assessment of residual bituminous coating for testing by the boiling water method, capable of both providing a discrete estimate comparable to the expert one and a more general integral estimate of the coating.

## 2. Related Works

It is noteworthy that quite a few researchers in the emulsion mixtures field provide automated visual assessment algorithms for residual adhesion after physical impact resistance testing. As a rule, such an algorithm is divided into two stages: recognizing stones in the analyzed image and semantically segmenting the stone surface.

The pixel-by-pixel filtering algorithm for selecting stones in the image is employed in all the papers reviewed, wherein there are variations in the filtering method used: some of them imply filtering regarding a stipulated background color value, whereas others ascertain the background color boundary by scrutinizing the color component histogram. As the rolling bottle test features in the majority of these studies, the surface is divided into two classes: bare gravel and bitumen layer.

Källén et al. [3] proposed a pixel filtering method for identifying stones in a picture. This method defines pixels with a small Euclidean distance value to white as part of the background and those with a large distance as stones. Subsequently, areas with identified stones are filtered to remove potential classification errors. Stones that are too small or possess significant differences between their area, as well as the ones with convex hulls, are excluded from the analysis. In general, in order to solve the problem of surface semantic segmentation, various approaches are utilized, from threshold filtering methods to deep convolutional networks. Let us consider some of them.

In order to segment the stone surfaces into zones of bituminous coatings and bare gravel, Källen’s team uses a graph-based segmentation algorithm, which includes the image being represented as a graph with pixels as nodes. Each pixel has a connection to each neighboring pixel, with a weight inversely proportional to the distance between these pixels in color space. Additionally, each pixel has an edge with nodes of typical bituminous coating and bare gravel colors. The edge weights are determined based on the proximity of the pixel color to the typical color. Typical colors, in turn, were determined through the clustering procedure by the k-means method on both completely bare stones and the ones completely covered with bitumen. The segmentation of the stone surface is performed through the procedure of removing faces from the graph until each pixel node of the image has a connection with only one node of a typical color while minimizing the sum of the weights of the removed faces.

In their further research [4] conducted in the same field, Källen’s group proposed a much simpler and faster segmentation algorithm. By means of capturing photos with varying illumination angles, the minimum and maximum illumination values for each pixel can be determined, and the difference between them can then be calculated. Two sets of stones, one completely covered with bitumen and the other completely bare, were used as samples to calculate histograms of distribution for intensity differences. These histograms were then used to derive the probabilistic intensity distribution for the two surface types. When analyzing the image of a stone based on these distributions, the probability of it having a bituminous coating is calculated for every pixel. The resulting quality assessment of the coating for the stone is calculated as the mean of these probabilities for all stone pixels.

Mulsow [5] employed akin methodology scrutinizing laser reflections from the stone surfaces in his publication. The stones were laser-illuminated both horizontally and vertically, and a composite image was then compiled. This image facilitated calculation of an illumination histogram used for establishing threshold values that discriminate between bare gravel and bituminous coating. Similarly, Källen et al. [6] investigated the distinction between natural illumination analysis and highlight analysis on stone surfaces. This approach is rooted in the fact that bitumen-coated surfaces tend to produce particularly strong highlights. Previous research provided the basis for their conceptual development.

The identification of bare gravel areas through pixel classifications according to their color characteristics is a commonly employed technique due to its straightforwardness. The ImageJ program for image processing is frequently applied in such scenarios. For instance, Blom and De Boek [7] used ImageJ for image binarization by Otsu’s method, in which the separation of two surface classes is made by the color intensity distribution histogram since a black-and-white image of a stone is considered. In a similar manner, Lantieri et al. [8] applied ImageJ to classify pixels according to their color characteristics in the YUV color scheme. The surface division into bitumen and gravel is made via pixel filtering through threshold values for each of the YUV’s components. Riekstins et al. [9] applied nearly the same approach involving the use of ImageJ. The key distinction of their research lies in the application of initial image processing methods, which imply enhancing hue, saturation, and brightness. This, in turn, amplifies the contrast between the gravel foundation and bitumen and increases the gap between them in the feature space, thereby simplifying the separation process.

Trejbal et al. [10] proposed an entropy segmentation algorithm to segregate bitumen-coated and uncoated stone zones. The algorithm evaluates surface smoothness using the image local entropy estimate. The algorithm consists of three steps: filtering by intensity, filtering by entropy, and smoothing. At the first step, the image is converted to grayscale, after which a primary threshold filtering of background is performed. To distinguish the pixels in areas with bitumen coating, it is necessary to take into account the texture of the area in which the pixel is located. To perform this step, the authors propose to calculate the local entropy for each pixel of the stone in a small radius neighborhood. Then, the obtained values are normalized based on the maximum entropy for the stone. After this step, threshold filtering is performed. At the final stage, the segmentation result is smoothed using a Gaussian filter to eliminate local errors. High-entropy areas indicate the bare gravel zone, whereas low-entropy areas indicate the bituminous coating zone.

Xiao et al. [11] described an algorithm to assess the residual adhesion of bitumen, which was specifically tailored for boiling water tests. The authors compared images of the gravel before and after the test procedure. In order to obtain a better assessment, an additional image sampling was applied: the color space of each RGB component in the image was represented as a union of consecutive non-overlapping intervals of a given width (the authors used an interval of width 20). For each pixel, the values of its color components were reduced to the mean value of the interval in which they fell. Thus, a significant reduction in the color heterogeneity of stone surface areas belonging to the same class was achieved, which simplified the segmentation procedure. Afterwards, the main color options characterizing the gravel base were determined from the histogram of the image of the original gravel. Accordingly, using these options, the number of pixels depicting bare surface in the image of boiled stones was determined, and the ratio of bare gravel pixels to the number of all stone pixels was calculated. Although this method was described specifically for the case of testing bitumen adhesion by the boiling method, the proposed approach deviated slightly from the surface segmentation methods using color characteristics that were applied in studies involving the rolling bottle technique. However, it did not take into account the factor of thinning the bitumen film, rather than its complete separation; as a result, the color characteristics of bare gravel could be greatly distorted.

In a study by A. Chomicz-Kowalska [12], residual adhesion was likewise explored after the boiling water method, utilizing conventional surface segmentation. Although the test stones were also segmented into two categories, the HSV color scheme was applied instead of the RGB scheme used in Xiao’s study. An additional distinctive characteristic of their methodology involved performing surface segmentation on both sides of the stones being studied, with the final values of surface area classes being calculated using images from both perspectives.

In the majority of the reviewed works that rely on threshold filtering, the thresholds themselves are set manually by the researchers based on their perception, which significantly reduces the objectivity of the method and the level of its automation. Moghaddam et al. [13] applied the most complex approach, avoiding this disadvantage. After preprocessing, the pixels in the image are grouped into clusters based on color similarity via the k-means algorithm. A class and the mean value of the members of this class are assigned to each pixel. Afterward, to determine whether each class belongs to bitumen or bare gravel, a classification algorithm (SVM or decision tree) is applied to their averages. For the training, average scores corresponding to the bitumen surface were prepared in advance, and the average values for bare gravel were taken from the images before the test. Thus, the algorithm needs to be pretrained for each run. This algorithm automates the coating evaluation process for the rolling bottle method. However, as it is entirely based on using the color characteristics of the gravel before testing and does not consider the surface relief, this method can be ineffective for the boiling method application.

In his work [14], ZhenZhou Wang proposes a universal algorithm for segmenting surface areas in an image based on the threshold filtering approach. In order to determine the threshold values by which the image is segmented, the derivative of the slope difference function of the intensity distribution is used. The original RGB image is converted to grayscale, and a normalized intensity histogram is calculated for it. The resulting histogram is then converted into frequency space via the Fourier transform method. The frequencies are filtered then to retain only the very low and very high frequencies. After frequency filtering, the histogram is inversely transformed back to the original intensity space. Then, the histogram slope to the right and left of each intensity value is calculated. This is performed by finding linear models through regression over *N* adjacent points on each side. The difference in the slope coefficients of the constructed models gives the value of the difference in slopes at a point. The extremum points are then found by calculating the derivative function. The points where the slope difference function reaches a local minimum are considered class separation points. They are selected as thresholds for segmentation. The same algorithm is applied to calculate threshold values for the image gradient map. Then, depending on the task, either segmentation by initial intensities, gradients, or a combination of both are used as the final segmentation map.

Such methods can be further improved, for example, by a finer selection of threshold values. In particular, ZhenZhou Wang et al. propose a method for flexible and robust threshold selection based on the slope difference distribution [15]. The threshold is selected based on the analysis of the peaks and valleys of the slope difference distribution. The pixel classes are separated according to the parameters calibrated for each specific image type. However, the method requires manual parameter adjustment and comparison with a benchmark obtained for each type of image being studied.

One of the key drawbacks of the algorithms reviewed is that they segment the surface into just two classes. While it is acceptable for the rolling bottle method, where the bitumen layer chips away, the process is unsuitable for boiling water tests, as their results do not correctly represent the observed coating quality. Furthermore, the pixel-by-pixel filtering relative to threshold values, along with the analysis of a single pixel on the surface, disregards the surface relief. As a result, this method lacks the ability to accurately identify areas with a thin coating layer. If it is necessary to take into account the relief of the analyzed surface and some more complex interconnections between neighboring pixels, it is advisable to employ more compound and advanced techniques of semantic image segmentation. In particular, such problems are well solved by deep learning methods, as shown by recent studies in the field of object detection and image segmentation [16,17,18,19,20,21]. Let us consider some state-of-the-art approaches.

A neural network with an encoder–decoder architecture is the dominant technique in the field of semantic segmentation [22]. This deep learning architecture comprises two major parts: an encoder and a decoder. The encoder converts the input data into a compressed representation, also known as a latent space. The decoder then generates the output data based on this compressed representation. The encoder and decoder are jointly trained using error backpropagation to minimize the gap between the predicted and actual outputs.

One of the key strengths of the CNN encoder–decoder model is its capability to process complex and multidimensional data, such as images and time series data. This is achieved by using convolutional layers in both the encoder and decoder to extract the relevant functions from the input data. Furthermore, employing latent space enables the model to apprehend the main data patterns and interconnections, which can prove useful for tasks like anomaly detection and data clustering.

The U-Net architecture is one of the most efficient and extensively utilized structures employing this technique. It was firstly introduced in the paper [23] by Ronneberger et al., in which its effectiveness for semantic segmentation in biomedical data is demonstrated.

The principal drawback of the basic encoder–decoder architecture is the problem of positional information loss when overcompressing, which subsequently results in a decline in segmentation quality during restoration to the original dimension. Therefore, the U-Net architecture is a significant innovation that incorporates direct connections between the encoder outputs and the decoder inputs at the same level. Firstly, this feature allows the decoder to use positional information when restoring dimensionality, and, secondly, it enables the usage of the convolution results and therefore the identified dependencies at every encoder level.

In [24], Enshaei et al. presented a study showcasing the effectiveness of the U-Net-like architecture in semantically segmenting material surface defects with varying material textures. They compared the efficiency of defect detection on a DAGM dataset of textured surfaces by two implementations of the U-Net architecture. The first model was the basic U-Net version with the only change in the order of the normalization and activation layers. In the second model, the max-pooling layer was replaced with a 3 × 3 convolution layer with strides 2 × 2. According to the test results, the second model showed slightly better results. It should be noted that this method is quite general and requires further development for application to bitumen analysis problems.

The U-Net architecture is also widely used in road surface image segmentation problems. For example, Majidifard et al. [25] utilize the U-Net network to determine the condition of pavements. To assess the deterioration of the asphalt, they perform localization of the defected road area by the YOLO model and then apply directly pixel-by-pixel segmentation of the defect by the U-Net model.

Similarly, Robet et al. [26] used a U-Net-based neural network to tackle the task of semantically segmenting the roadway for pavement type determination and defect detection. The specialty of the approach is that images for analysis are captured by a road surveillance camera (the Road Traversing Knowledge (RTK) Dataset is used for training). The neural network trained on this dataset made it possible not only to detect the road surface in the image but also to determine its type (asphalt, paved, and dirt) and to identify elements on the road such as markings, tire tracks, patches, potholes, and puddles.

However, in addition to the neural network approach, other machine learning techniques can be used in the task of semantic segmentation, which, due to greater simplicity and a smaller number of parameters, allow one to achieve relatively good results in less computing time [27,28].

One of these techniques is gradient boosting. It is an ensemble learning method that combines several weak models to create a general model capable of describing complex dependencies. The main intention of gradient boosting is to iteratively add new models to the ensemble, whereas each new model corrects the errors of the previous ones. The use of gradients, which are mathematical functions that describe how the model’s predictions should be adjusted to minimize the total error, is crucial to this approach. At each iteration, the gradient boosting algorithm first calculates the loss function gradient in relation to the predictions of the current model. It then establishes the appropriate direction in which to adjust the model’s predictions to minimize the error. The new model is adjusted to the anti-gradient using a decision tree, correcting the previous model’s errors. The new model is then added to the ensemble, and the process is repeated until the stop criterion is met.

The effectiveness of the gradient boosting algorithm is demonstrated by Dai et al. in their paper [29], wherein they successfully solve the task of detecting an object in an image. To achieve this goal, they perform several convolutional transformations with different kernels over the original image, thus forming a matrix of additional features. The gradient boosting algorithm (XGBoost) is then utilized to classify each image pixel as belonging to either the background or target object. The advantage of using XGBoost is that gradient boosting is more resistant to overfitting on small samples.

The algorithms considered provided the basis for the software component of the device for automated quality assessment of residual bituminous coating.

Given the applied nature of the present research, we did not strive to compare all known approaches to the image segmentation problem. Based on the analysis of modern surveys [19,20,21,22], we have chosen the Fully Convolutional Networks (FCN) approach as the most appropriate for the described problem. One of the most popular, relatively simple, and effective options for the FCN architecture is the U-Net architecture, which was chosen to implement semantic segmentation. In order to extend the study, XGBoost was chosen as an alternative approach with sufficient generalization capability. For comparison with “traditional” techniques of image segmentation, we implemented an algorithm based on threshold histogram analysis as the most common approach in the field of residual adhesion analysis.

Our contribution is as follows. We have developed an automated bituminous coating quality assessment device that provides the required reproducible conditions for image shooting and analysis. This eliminates the influence of light situation changes when analyzing samples. As well, we have compared the two most common approaches to image segmentation and identified their features with regard to the considered application task. The most suitable segmentation method is refined and incorporated into the industrial laboratory prototype.

## 3. Materials and Methods

### 3.1. Hardware and Software System Structure

As part of the research, a hardware and software system (HSS) was developed to perform the basic determination of the adhesion level of bitumen to gravel, as shown in Figure 2.

The developed HSS has the following characteristics. The device has an aluminum h519 × 394 × 445 mm frame (1). It is equipped with an internal cavity (2) for gravel samples to be analyzed. On the front side of the device, there is a sliding tray for the analyzed sample placement (3). The sliding tray has a hole for fixing the substrate (4), where the loaded samples are located. In order to operate this device, the power button (5) and a color LCD display (6) are located on the HSS’s front side. An Ethernet port on the back of the device for connecting to a PC acts as a setup and debugging interface (7). A USB camera (8) and a set of eight LED sources (9) are installed in the internal cavity of the device to analyze the adhesion level of bitumen to a stone. An Nvidia Jetson Nano microcomputer (10) is provided in the device to control the equipment and analyze samples. A three-pin voltage supply connector is located on the back panel to power the HSS (11). In the sample tray, there is a substrate marked, as shown in Figure 3.

The samples are put in the zone of the largest circle, which is the sample location zone. To be correctly processed, the samples must be placed as follows: the distance between the samples should not be less than 5mm, and they should not cross the line of the sample location zone or be located outside this zone. To put the samples into the device, the substrate containing the samples must be placed within a designated slot in a specially designed tray.

The device has several operational stages:Running a program: At this stage, the program displays a welcome message and waits for the analysis start signal from the operator;Hardware verification: At this stage, the program checks for camera connected to the device and, when it is detected, proceeds to shooting images. If, after fifteen seconds from the beginning of the transition to this stage, the camera is not detected, the program proceeds to the results stage, displaying the camera absence error;Shooting images: At this stage, the LED lighting is switched on alternately, and the pictures of gravel samples put into the HSS are taken. Upon successful completion of shooting at all illumination angles, the images are transferred to the processing stage. If an error occurs during shooting, the transition to the results stage occurs with the display of the shooting error;Processing: This stage involves processing and analysis of the obtained images of gravel samples in accordance with the specified requirements. If an error occurs, the corresponding error message is transmitted to the results stage instead of the analysis values;Results: The stage includes displaying on the HSS’s screen the obtained analysis values and images of stone samples in the outline in case of successful completion of the analysis. If an error occurs at one of the described software operation stages, the corresponding message is displayed on the HSS’s screen. In any case, this stage displays a button to start a new analysis;Device shutdown: The device shutdown occurs when the power button is held for several seconds while the device is operational. Its primary purpose is to confirm the operator’s intention to shut down the device. In case of confirmation, the main program is stopped, followed by the device logout and shutdown. Otherwise, there is a return to the executed stage of the main program.

The following requirements are imposed on the software component of the HSS:Input data should comprise images of gravel samples within the inner cavity, captured at various illumination angles;The software must determine the total number of stone samples present in the input data;For each found sample, the contours of the borders and a delineative rectangle should be defined;The adhesion level of each found sample should be analyzed independently;The analysis results should be presented in both integral and discrete forms, using a value that is applicable to all samples;In addition to the general values, the software tool should provide an image of the analyzed stone samples in a special color outline.

### 3.2. Software Component Overall Structure

Consider the implementation of the HSS’s software component. It is represented by a shooting module and an algorithm for assessing adhesion level. When the hardware system is run, and the program is loaded, the user puts the tested sample of stones into the device and runs the program through the user interface. Once the program is run, the shooting module captures a series of eight photos of samples at different illumination angles. The resulting images are processed by an adhesion assessment algorithm.

The adhesion assessment algorithm includes a stone detection module, a semantic segmentation module, and a final classification module. These modules are run sequentially and perform all the required image analysis steps (see Figure 4).

Upon completion of the final classification module, each stone from the test sample is assigned a discrete and integral (percentage) estimate. Then, the average integral and median discrete estimates for the sample are calculated. Afterwards, the resulting values are delivered to the user as final estimates of the adhesion quality in this sample.

The flow diagram for software operation is presented in Figure 5. The device is turned on by briefly pressing the button on the front side. After the system starts, the operator can select the analysis method (XGBoost or U-Net). Before starting the analysis, the operator must open the loading tray, fix the substrate, place the gravel samples on it, and then close the tray. To start the analysis, the operator should press the button on the LCD display, and then the camera launches. If the camera is connected successfully, the analysis is carried out automatically. The progress of the analysis is shown on the display. Once the analysis is complete, the results obtained will be displayed. In case of an error, the corresponding message will be shown. To shut down the device, the operator should hold down the power button for a couple of seconds.

### 3.3. Stone Detection

The stone detection module is responsible for detecting all the stones in a photo series to form a mask and a delineative square for each stone, followed by cutting it out of the overall image.

Classical algorithms of the Canny operator [30] and Suzuki contour search [31] are used for this task.

The Canny Edge Detection Algorithm is a widely applied boundary detection method in computer vision and image processing. The algorithm comprises four main stages: smoothing the image with a Gaussian filter, calculating the magnitude and gradient orientation, suppressing non-maximal boundaries, and applying hysteresis threshold screening to obtain the final boundary map.

A known limitation of the Canny algorithm is its sensitivity to color intensity gradients, often leading to blurring in color areas. Shadowed regions, where the stone casts a shadow, show black as the dominant color, thereby triggering blurring. To remove shadows from the image required for stone detection, the maximum value for each pixel across all images is calculated (see Figure 6).

To improve the Canny filter operation, contrast enhancement is applied to the maximum image in order to increase the visibility of the blue component characterizing the background, followed by the removal of all pixels with a dominant, relatively large blue component (see Figure 7).

After these operations, Canny easily finds all the necessary stone boundaries (see Figure 8).

Suzuki’s algorithm is a contour detection algorithm based on the Freeman chain code used in computer vision and image processing, which represents the boundary of an object as a sequence of directions. The algorithm first identifies the starting point of the contour and then uses the Freeman chain code to follow the object’s boundary. The algorithm uses a stack to track the current position and direction of the contour and an array of visits to mark pixels already visited. If the algorithm encounters a pixel that has already been visited, it returns until it finds an unvisited pixel and continues to follow the boundary from there. When the contour is completely delineated, it is added to the list of contours.

As the algorithm does not provide any requirements for the number of stones in the analyzed set, all possible contours are initially selected. Then, a contour with a maximum area is selected, relative to which all other contours are filtered. A contour is considered a contour of a stone if its area is at least 20% of the area of the largest stone. All other contours are regarded as dirt on the substrate.

After filling in all the remaining contours, we get the resulting masks of stones (see Figure 9).

Then, a delineative square is determined for each stone by its mask, and this part of the image is cut out of each photo. This set of images of the stone with the mask is passed on to the semantic segmentation module.

### 3.4. Preparing a Dataset for Model Training

Training a neural network to achieve a high-quality result requires a sufficiently large set of qualitatively marked up images in the training sample. Let us further consider preparing a dataset for training and a neural network.

In this research, we use a dataset consisting of 300 stones. Each stone’s discrete estimate was determined by an expert, and half of them were additionally categorized by the surface types indicated in Figure 10. The surface types include areas of bare gravel indicated in red, insufficient layers indicated in green, and thin layers indicated in blue. Moreover, areas of the stone lacking a color mask were defined as the ones having a thick layer.

Of the set of marked up stones, 70% was allocated as a training sample for the segmentation algorithm, and the remaining 30% was allocated for a test sample. In such a way, we get a fairly limited sample for training a deep neural network, which can easily lead to the model retraining.

Accordingly, as an alternative approach, it makes sense to try a simpler machine learning algorithm that would rely on the experience of existing research’s algorithms, in which image segmentation primarily relied on the color characteristics of individual pixels. To study the difference between the surface types under consideration (thick layer, thin layer, insufficient layer, and bare gravel) in the color space, we construct distribution histograms of color characteristics of stones across all images with a marked-up surface.

In order to view the image of the stone without strong distortions that arise due to the illumination angle, 8 photos with different illumination angles are taken for each stone. To eliminate the influence of excessive illumination, highlights, and shadowing, an aggregated image is applied in the analysis. As the highlight on the bitumen surface introduces too much bias into the average value, instead of simple averaging, 30% of the percentile is used to obtain an aggregated image. The image obtained by this aggregation method will hereinafter be referred to as the average image.

## 4. Experiment Results

### 4.1. Analyzing the Color Components in the Image

We consider the distribution of color components in the three most common color spaces, specifically RGB (Figure 11), HSV (Figure 12), and YUV (Figure 13). If there are significant differences in the distributions of color components for various surfaces, we can conclude that the conditional classification by color components is sufficiently effective and also determines the best color spaces to use.

Based on histogram analysis, it can be concluded that each of the four surface types has a distinctive distribution pattern for color characteristics. This is apparent mostly in the HSV color scheme and least so in YUV. Intersections in the distributions are normal since, with a sufficiently rough markup, zone parts of one type can affect zone parts of another type. One of the most challenging aspects of intersections is the insufficient coating layer since it very often delimits areas with another coating type.

It is worth examining the characteristics of the color component distributions in the HSV space for various surface types, namely, the values of the median, lower, and upper quartiles (see Table 1). These values were obtained by averaging the values of a sample of test images for samples of clearly distinguishable class of bitumen coating (see Figure 12).

Table 1 shows that the value component is one of the most effective criteria for separation. On the whole, we can conclude that it is possible identify suitable threshold values for components, resulting in the satisfactory separation of sections with differing surface types.

However, to attain higher quality segmentation, the algorithm should consider more intricate pixel color dependencies with neighboring pixels, as well as the relief and highlights of the surface. In this instance, employing a gradient boosting algorithm represents the optimal choice.

In order to select a suitable method for assessing the quality of bitumen coating, we implemented three semantic segmentation methods, namely, threshold segmentation, gradient boosting, and a U-Net-based neural network.

### 4.2. Gradient Boosting

According to the stone images received at the module input, a dataset of feature vectors is formed, in which each vector corresponds to a specific pixel of the stone surface. The features used for classification are presented in Table 2.

The gradient boosting algorithm implementation employed XGBoost. The model was trained by combining datasets of feature vectors from all stones in the training set into a single large dataset. A 10% validation sample was used to determine the best model by selecting optimal hyperparameters through random search.

Random search grid parameters:Tree depth in the [3, 14] interval;Number of trees in the ensemble in the [50, 500] interval;Training parameter: [0.1, 0.05, 0.01, 0.005, 0.001];Tree subsample proportion: [1, 0.75, 0.5, 0.25];Tree features proportion: [1, 0.75, 0.5, 0.25].

Best model parameters:Tree depth: 12;Number of trees in the ensemble: 480;Training parameter: 0.1;Subsample proportion: 0.75;Features proportion: 1;Validation accuracy: 93.43%.

After classifying each pixel of the stone, a primary map of the surface type distribution is obtained (see Figure 14a). To neutralize possible errors and smooth out the zone limits, a median filter with a kernel (7, 7) is applied to this map (see Figure 14b).

### 4.3. U-Net Architecture

Unlike gradient boosting, where the model analyzes the image pixel by pixel so that it considers only the pixels belonging to the stone, a rectangular matrix must be input to the convolutional neural network, i.e., in addition to distinguishing four surface types, the neural network must also determine the background in the image. The average image is used for the analysis.

The standard approach for semantic segmentation of different dimension images using neural networks is to reduce images to the same size. However, in this case, a different approach involves splitting the original image into tiles of the same small size, segmenting all the tiles and then merging them into an image of the original size. This approach was chosen in order to increase the training sample size to achieve better model generalization.

For the training dataset of stones, the average image sizes are (376, 403), so (128, 128) was chosen as the one tile size. Firstly, this tile size enables the use of a neural network with a relatively small number of parameters, and, secondly, it allows to significantly increase the training sample size through synthetic instances obtained through rotation operations, whereas the tile size is large enough to capture patterns of textures and relief of various surface types.

As part of the search for the neural network optimal configuration, the following parameters varied:Compression depth: [2, 5];Base number of filters: [2, 16];Number of convolution blocks at the level: [1, 3];Kernel size in the convolution block at the level: [2, 6];Activation function used in the convolution block: [ReLU, Leaky ReLU, ELU, Sigmoid];Max pooling kernel size: [2, 4];Training parameter: [0.0001, 0.1] with logarithmic progression.

In order to determine the optimal network configuration, a Bayesian optimization algorithm was applied with a maximum number of 800 tested models and with 200 starting points (see Figure 15).

MSE was used as a loss function to ascertain that not only does the surface type match its true value but also that the falsely defined class is as close to the authentic one as possible when misclassified.

To optimize the parameters, we applied the Adam algorithm with 50 epochs, a validation sample size of 10%, the possibility to initiate early stopping in the absence of MAE metric improvement during validation, and a patience parameter of 5 epochs (see Table 3).

For the ten models with the highest quality obtained through Bayesian optimization, further training has been carried out for an extra 1000 epochs, with a decrease in the training parameter and the possibility of an early stopping with a patience parameter of 10 epochs.

At the output of the neural network, we get an assessment matrix of surface classes for pixels in the tile. As the estimates obtained are real, in order to obtain a map of the surface type distribution, after the tile merging procedure, the procedure of rounding values to integers is applied to the complete matrix, as well as the truncation of estimates in the [0, 4] interval.

The final map is then smoothed using the median filter (7, 7), similar to the case of gradient boosting (see Figure 16).

### 4.4. Comparison of Segmentation Accuracy

In order to perform an unbiased image segmentation quality assessment, the results of the developed models have been compared with the accuracy in naive segmentation. During this process, the entire stone surface is determined by one surface type for all the stones in the test sample (see Table 4 and Table 5). Metrics utilized include the average segmentation accuracy of the test sample and the average absolute classification error.

Also, a simple and quick semantic segmentation method was implemented to compare the results obtained with a widely used threshold segmentation approach (TS). It is based on filtering pixels by a threshold value and is used in the majority of the reviewed works. Therefore, we adapted the algorithm for image segmentation for all surface types under consideration. In addition, we applied a number of known effective practices: image analysis in the HSV [12], in which surface types are best separated, and the use of additional image sampling [11], in order to simplify the task of component filtering of image pixels. The implemented approach includes the following steps:Converting all frames from RGB to HSV;Determining the mean of the image for H and V components;Sampling the image into areas of width 20;Partitioning the stone surface into regions where gravel surface is not detected (class 1) and where gravel surface is detected (class 2) based on comparison with threshold values;Partitioning the regions of class 1 into areas with insufficient coverage and bare gravel;Partitioning the regions of class 2 into areas with a thick and thin layer of coating.

The threshold values of H and V components were determined on a training set of images with a marked surface. For each stone, the optimal values for dividing the zone were determined according to the rule described above from the point of view of maximizing the Kullback–Leibler divergence. Then, the obtained threshold values were averaged over all stones.

It is apparent that both XGboost and U-Net demonstrate comparable results, significantly better than naive segmentation and TS. The U-Net shows marginally superior precision in identifying surface types, but XGBoost deals with surface type classification errors better than U-Net.

### 4.5. Final Classification Module

Having passed the semantic segmentation module, the final distribution map of the stone surface types is transferred to the final classification module. Here, the areas of each surface type are calculated in relation to the total area of the stone.

The integrated (percentage) quality assessment of the stone coating is calculated as a weighted sum of data:*R* = (0.2*Weak* + 0.5*Thin* + *Thick*) × 100,(1)
where *R* is an integral estimate, *Weak* is an insufficient layer relative area, *Thin* is a thin layer relative area, and *Thick* is a thick layer relative area.

For identifying the discrete estimate of the stone coating, in order to simulate the expert’s logic while determining the estimate and prevent retraining, a classifier based on a decision tree is used, with a maximum depth of 5 and with a minimum number of 7 elements in the list. The features in the classification of a stone are its values relative to the surface type areas.

All stones with marked up surfaces are utilized to train the classifier for discrete estimates.

The final test studied the correspondence of discrete estimates of stones determined by the algorithm to expert’s discrete estimates for all stones in the sample. The results obtained using the XGBoost and U-Net segmentation models, as well as threshold segmentation, are compared (see Table 6).

As can be seen from the results, the accuracy of a simple threshold segmentation is significantly lower. Approximately in 8.52% of cases, the TS method assigns a stone to a different class relative to expert assessment. The main disadvantage of this algorithm is that, due to the classification of an isolated pixel, it disregards the surface topography. Therefore, it is not able to qualitatively distinguish the class 5 and class 4 (among the errors with overestimation of stone, the share of class 4 is 53%). In both cases, the surface of the stone is completely covered with bitumen, but there is a difference in layer thickness. This can be significant when industrially testing samples of asphalt mixtures.

Assessing the quality of the developed solution, it can be concluded that when using the XGBoost-based segmentation model, the algorithm shows better results, compared with U-Net, although the difference in quality is negligible. The XGBoost-based model is more likely to overestimate the stone value, whereas U-Net tends to underestimate it more frequently.

Generally, the algorithm’s estimate coincided with the expert’s one in over 70% of cases. In an additional 22% of cases, the algorithm’s estimate differed from the expert’s by only 0.5 points, which falls within the threshold value. The algorithm’s estimate differed from the expert’s one by more than 1 point in only 2% of cases, with the maximum error of 1.5 points. Therefore, a stone with an expert estimate of “5” has never received an algorithm estimate of “3” or lower, and vice versa.

### 4.6. Performance Analysis

During image processing, the algorithm combines the received images into one, segments the bitumen samples in the combined image, and processes each region of the segmented samples individually. As the developed HSS has restrictions on the hardware installed, the running time of the algorithm varies depending on the properties of the set of samples under analysis. To reveal dependencies between the algorithm’s performance and the parameters of the samples, some tests on gravel of various numbers and sizes have been carried out. For the experiment, sets of 1, 3, and 5 gravel samples were used. For each sample size, five sets of available samples were applied. Before running tests for a new sample size, the application was restarted. Also, for each sample, the experiment was averaged over three runs. For the test, random samples of various sizes were used (see Figure 17). The average operating times of the algorithm are given in Table 7.

The results show that the operation of combining images and searching for stone masks takes approximately the same time. However, due to the performance degradation after initializing the XGBoost model on the prototype, there was an increase in processing time. In addition, the impact of the initialization of the XGBoost model on the execution time of the algorithm is noticeable. The experiment showed a great variability in the time of individual processing of stones in the sample: from 22 s to 75.9 s. It should be noted that as the number of samples in a set increase, the processing time for an individual sample decreases slightly due to the peculiarities of initialization and testing of the model. Analysis execution time is affected by throttling of the device’s CPU after the XGBoost model is initialized.

## 5. Conclusions and Further Development

The adhesion properties of bitumen determine the quality and durability of road surfaces. According to existing practices, they are determined subjectively by experts, which entails known disadvantages. The aim of this work was to develop a methodology able to replace the human in assessment of bitumen quality regardless of the conditions under which the analysis is performed. For this purpose, hardware that “works out of the box” was developed, facilitating sample processing without requiring preliminary preparation and configuration.

To ensure the assessment accuracy, two versions of semantic surface segmentation algorithms have been developed by the authors. After comparing the results, it is evident that the gradient boosting approach outperforms the U-Net architecture for the purposes of the research. In general, both algorithms show a fairly high coherence of their estimates with expert discrete estimates. The significant discrepancy (error in determining the class of the sample) with the expert’s assessment was less than 2%.

A possible explanation for the reason why the U-Net architecture has shown worse quality compared to the gradient boosting method is the insufficiency of the training sample, from which the model fails to extract general patterns. Therefore, the most straightforward and evident direction for further development is to increase the number of marked up samples.

Another important direction of research is the reduction in image processing time. As this work is devoted to a laboratory prototype, the main requirement for it is the accuracy of assessing the quality of bitumen coating, expressed in approximation to expert assessment. At the same time, sample processing time was not a key requirement. With further development of the proposed methodology, a conveyer analyzer can be implemented with greater computing resources, parallelization of calculations, and optimization of calculation time.

It should be taken into account that the thickness of the covering bitumen layer is a significant factor in the expert evaluation, against which the accuracy of the developed algorithms was assessed. A significant contribution of our work compared to binary classification of the known methods is that the developed solution is capable of classifying the gravel samples considering the thickness of the covering bitumen layer. It is also worth noting the general differences, both in terms of hardware implementation (number of images, shooting angle, and distance to the lens) and in the methodological part (calculating the accuracy of clustering for all stones in the image or for each one separately and the number of stones in the image, as precise surface classes have been formed). Unification of these conditions allows us to implement a more sophisticated methodology that makes our developments applicable at the industrial level.

The authors have developed and registered a patent for a hardware and software system and a methodology for automated assessment of the adhesion of bitumen to gravel. These developments may further lead to the establishment of pavement durability and quality standards. Our methodology minimizes testing errors and can be used to create a database of bituminous mixtures tests. In addition, the developed methodology allows standardizing the data obtained and contributes to the creation of digital research database compiled of various sources. If the HSS application is successful, we will consider its launch into mass production. Incorporating the developed solutions into existing adhesion level test standards may result in improved primary tests for asphalt quality.

## Figures and Tables

**Figure 1 sensors-23-09325-f001:**
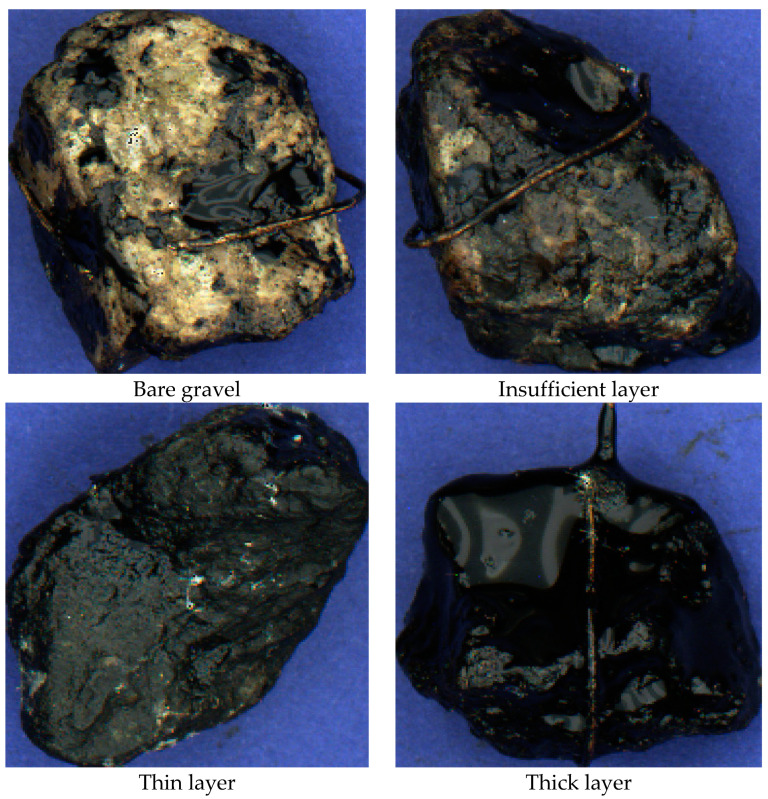
Gravel surface coating types. Average stone size: 3 × 2 cm. Average stone image size: 378 × 402 pt.

**Figure 2 sensors-23-09325-f002:**
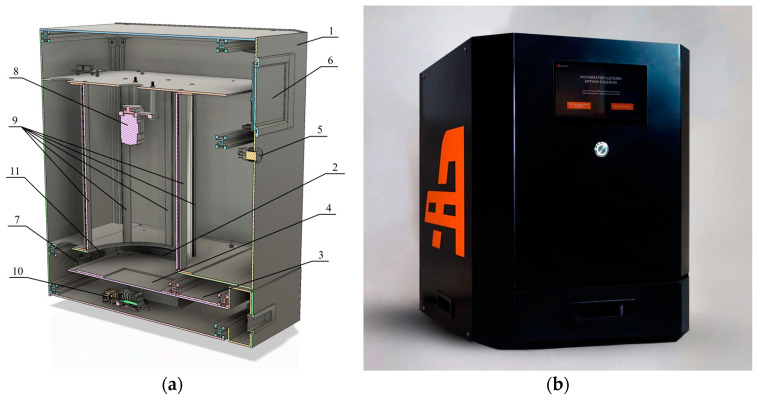
(**a**) Device diagram; (**b**) Implemented prototype.

**Figure 3 sensors-23-09325-f003:**
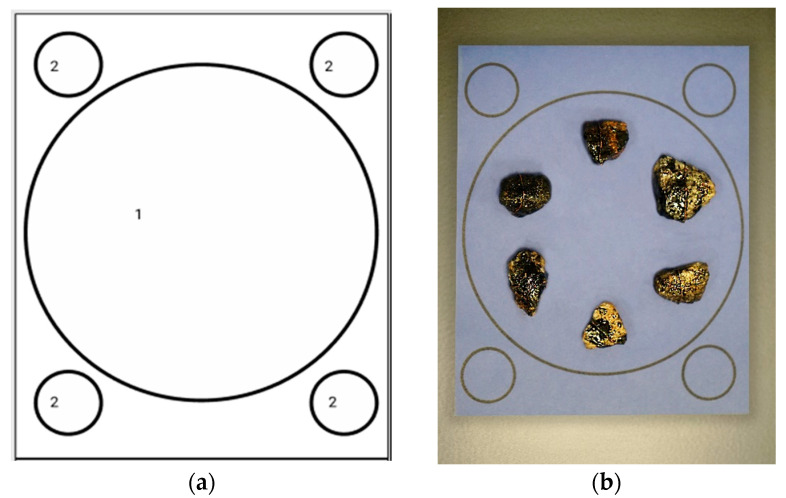
(**a**) Substrate template, where 1 is the area for the samples, and 2 is the area for color adjustment; (**b**) Samples placed on the substrate.

**Figure 4 sensors-23-09325-f004:**
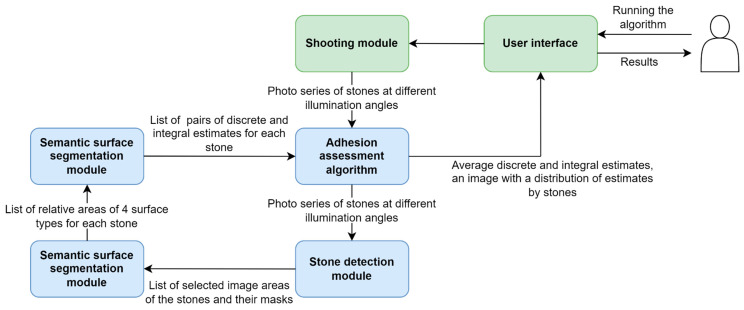
Hardware system overall operation scheme.

**Figure 5 sensors-23-09325-f005:**
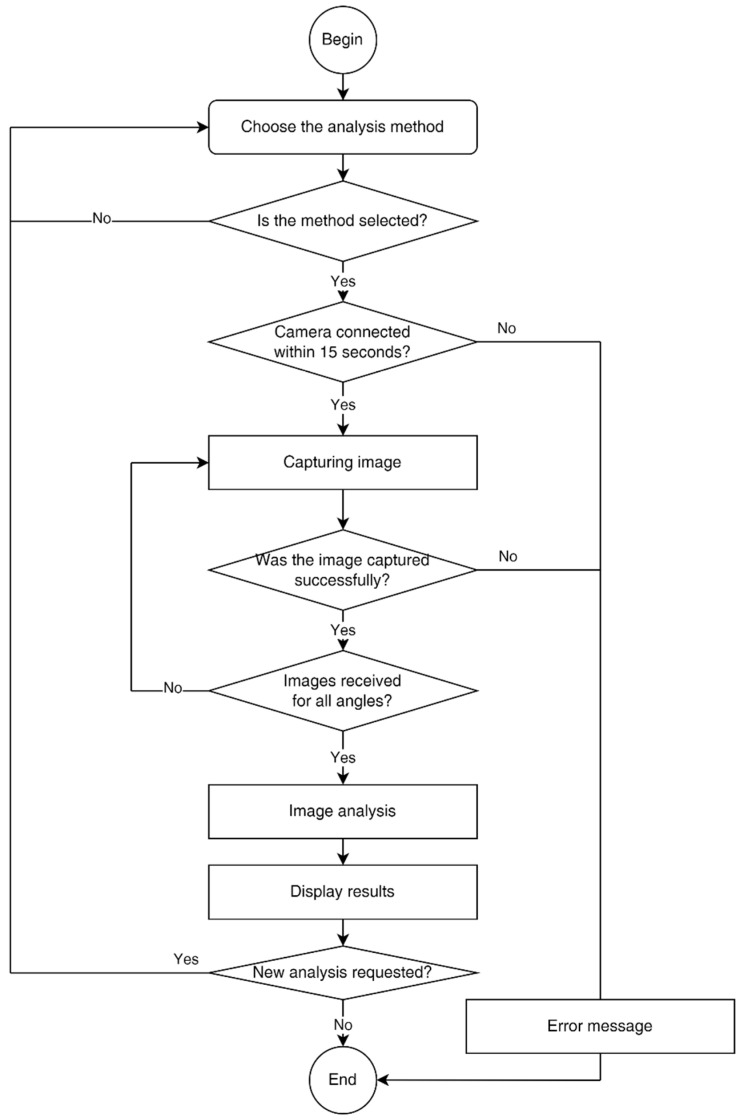
The flow diagram for software operation.

**Figure 6 sensors-23-09325-f006:**
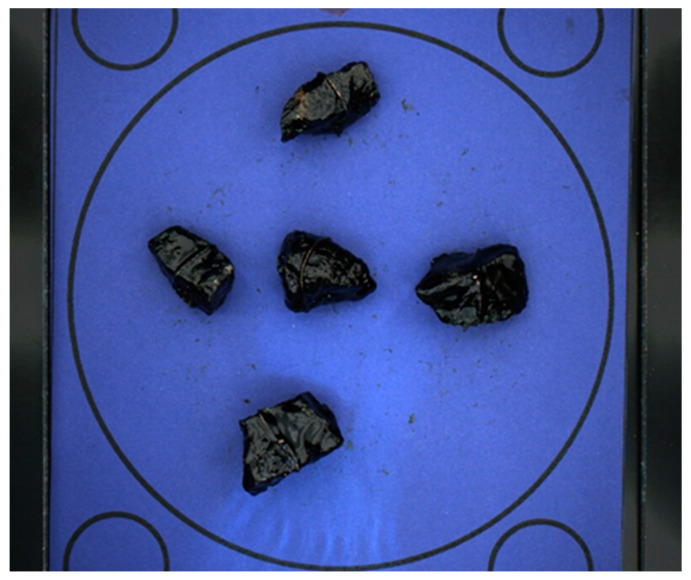
Example of the maximum for all frames.

**Figure 7 sensors-23-09325-f007:**
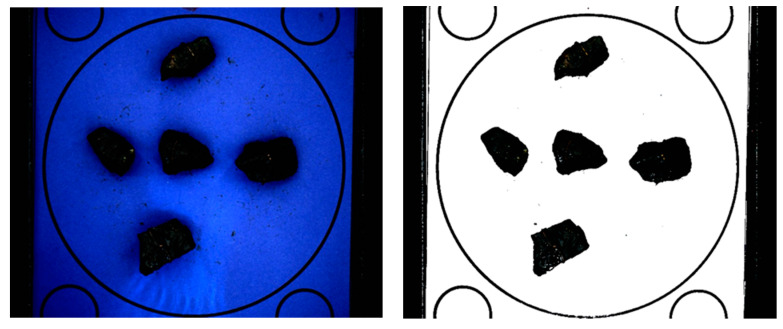
Increased contrast and removed blue background.

**Figure 8 sensors-23-09325-f008:**
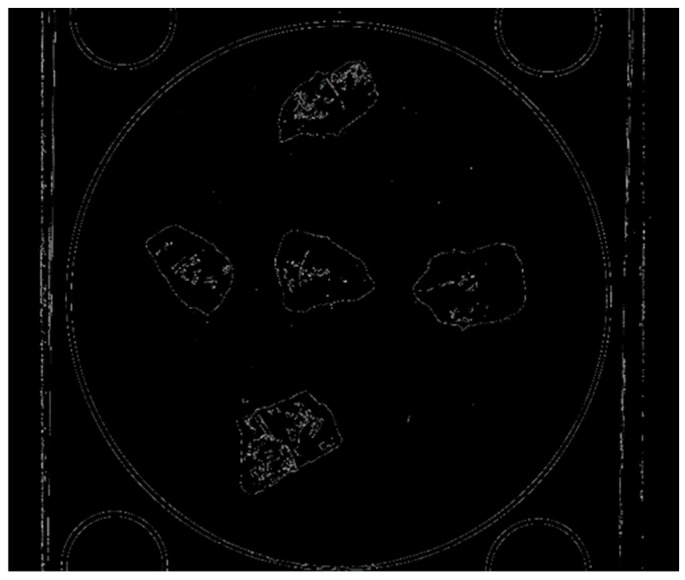
The contour search algorithm is used for the boundaries obtained via Canny operator.

**Figure 9 sensors-23-09325-f009:**
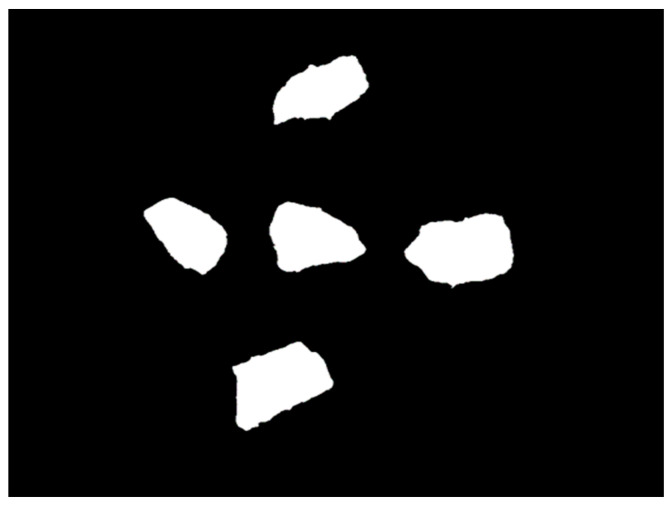
Resulting masks of stones.

**Figure 10 sensors-23-09325-f010:**
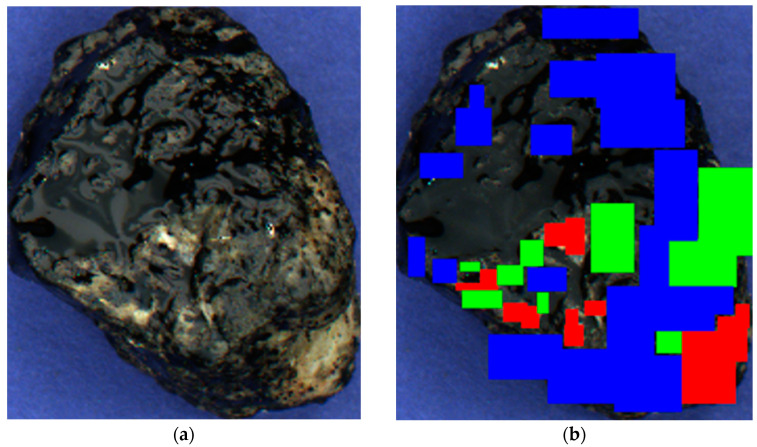
Stone markup by surface types: (**a**) The original photo; (**b**) The marked-up image. Each color represents regions belonging to the same class.

**Figure 11 sensors-23-09325-f011:**
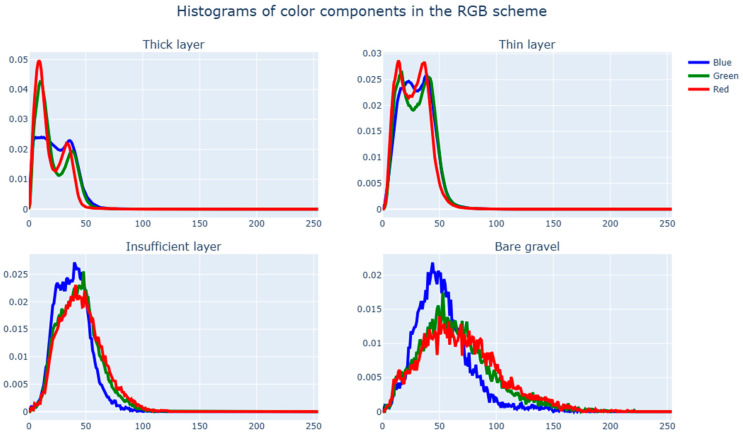
Histograms of color components in the RGB scheme.

**Figure 12 sensors-23-09325-f012:**
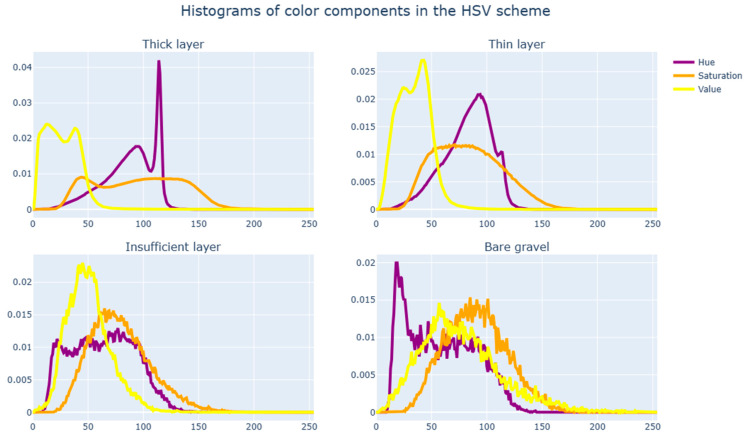
Histograms of color components in the HSV scheme.

**Figure 13 sensors-23-09325-f013:**
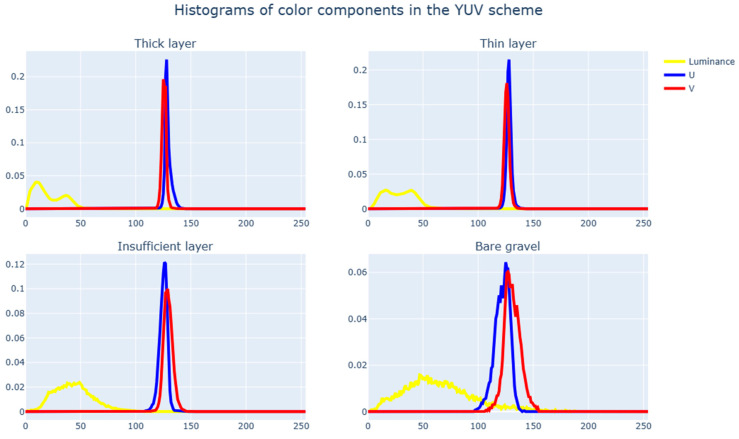
Histograms of color components in the YUV scheme.

**Figure 14 sensors-23-09325-f014:**
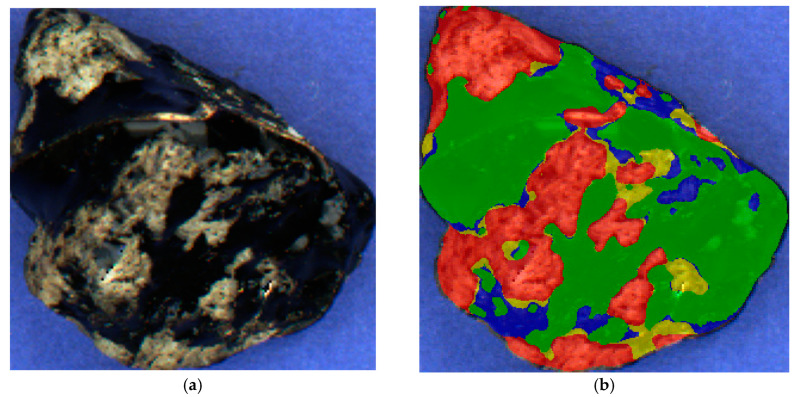
The average image (**a**) and the final map (**b**) of the surface type distribution, where red is bare gravel, yellow is an insufficient layer, blue is a thin layer, and green is a thick layer.

**Figure 15 sensors-23-09325-f015:**
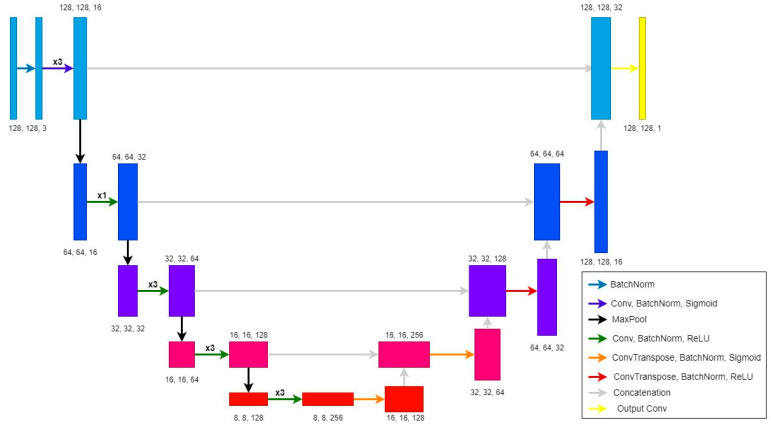
The final neural network architecture with the optimal configuration found.

**Figure 16 sensors-23-09325-f016:**
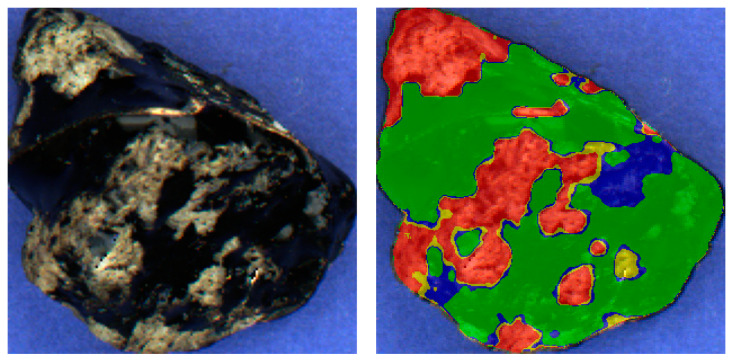
The average image and the final map of surface type distribution. Green color indicates thick coating; red—bare gravel; yellow—insufficient layer; blue—thin layer.

**Figure 17 sensors-23-09325-f017:**
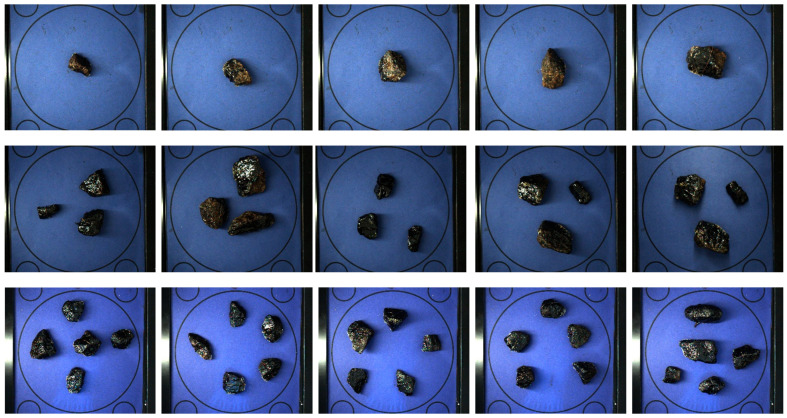
Sets of samples.

**Table 1 sensors-23-09325-t001:** Statistical characteristics of color component distributions in HSV space for different surface types.

Surface Type	Lower Quartile	Median	Upper Quartile
Bare gravel	29, 70, 51	56, 90, 72	84, 109, 96
Insufficient layer	41, 59, 37	64, 76, 49	86, 96, 62
Thin layer	70, 59, 23	86, 81, 34	98, 104, 44
Thick layer	74, 62, 14	92, 97, 26	108, 127, 38

**Table 2 sensors-23-09325-t002:** The features used for classification in gradient boosting.

Feature	RGB	HSV
Value in the average image	+	+
Average value of the average image around the (15, 15) and (21, 21) pixels	+	+
Median value of the average image around the (15, 15) and (21, 21) pixels	+	+
Value in the minimum image	+	+
Average value of the minimum image around the (15, 15) pixel	+	+
Median value of the minimum image around the (15, 15) pixel	+	+
Standard deviation of the pixel component for all frames	+	+
Average value of the standard deviation for all frames around the (15, 15) and (21, 21) pixels	+	+
Standard deviation value of the component of the average image around the (15, 15) and (21, 21) pixels	+	
Maximum value of the average image around the (11, 11) pixel	+	

**Table 3 sensors-23-09325-t003:** The resultant error of the optimal found model in training.

Sample	MSE	MAE
Training	0.0307	0.0534
Validation	0.1112	0.1074

**Table 4 sensors-23-09325-t004:** Naive segmentation results.

Surface Type	Average Accuracy, %	Average MAE
Bare gravel	1.86	2.41
Insufficient layer	14.36	1.45
Thin layer	24.86	0.77
Thick layer	58.92	0.59

**Table 5 sensors-23-09325-t005:** Model segmentation results.

Model	Average Accuracy, %	Average MAE
TS	49.79	0.68
XGboost	77.03	0.26
U-Net	77.04	0.27

**Table 6 sensors-23-09325-t006:** Comparison of test results.

Metric	TS, %	XGboost, %	U-Net, %
Accuracy	62.96	75.56	73.7
Estimates less than the expert one by 0.5	12.59	12.22	15.19
Estimates greater than the expert one by 0.5	15.19	9.63	7.78
Estimates less than the expert one by 1	2.96	1.11	0.74
Estimates greater than the expert one by 1	5.56	0.74	1.11
Estimates that differ from the expert one by more than 1	0.74	0.74	1.48
MAE	0.239	0.139	0.156
Maximum deviation of the estimate	2.5	1.5	1.5

**Table 7 sensors-23-09325-t007:** Average processing time.

Set Size	Combining Images, s	Search for Stone Masks, Average by Sample, s	Surface Analysis, Average by Sample, s	Total Time, s
1	0.8	3.62	59.28	63.88
3	0.8	3.22	56.09	172.74
5	0.8	3.18	38.94	198.92

## Data Availability

No new data were created or analyzed in this study. Data sharing is not applicable to this article.

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
