# Peer review of "Design and Implementation of a Hardware and Software System for Visual Assessment of Bituminous Coating Quality"

_sensors, 2023, doi:10.3390/s23239325_

Round 1
Reviewer 1 Report
Comments and Suggestions for Authors
This paper introduces a hardware and software system for automating the visual assessment process of residual coating on stones, providing accurate results to ensure road durability and quality standards. The system can perform evaluations under static conditions and automatically create a database of bituminous mixture tests, reducing testing errors. To improve the paper's quality, the following comment needs to be considered.
1. The sentence is too long and is not easy to read, please carefully check the whole manuscript, for example:
- Page 4, Lines 157-175, Please integrate and improve these sentences to make the related work section easier to read.
- Pages 5-6, Lines 220-228, Please integrate and improve these sentences.
2. Page 5, Line 240, (Figure 3)--> as shown in Figure 3
3. How about the computation time for the proposed approach
Author Response
Dear Reviewer,
thank you for your valuable comments. We carefully studied each of them and tried to take them into account in the manuscript. Changes (except minor changes and grammar) are highlighted in green.
- The sentence is too long and is not easy to read, please carefully check the whole manuscript, for example:
- Page 4, Lines 157-175, Please integrate and improve these sentences to make the related work section easier to read.
- Pages 5-6, Lines 220-228, Please integrate and improve these sentences.
- Page 5, Line 240, (Figure 3)--> as shown in Figure 3
We have rewritten these sentences.
- How about the computation time for the proposed approach
We have added a section 4.6. Performance analysis.
Thank you for your attention and collaboration! We hope the changes made have significantly improved the structure and readability of the manuscript. We welcome your comments on the updated version.
Вest regards,
the authors.
Reviewer 2 Report
Comments and Suggestions for Authors
I have the foolowing suggestions:
1, More image segmentation methods should be tested and compared with the used one. For instance, the following methods:
https://www.mathworks.com/matlabcentral/fileexchange/56371-zhenzhou-threshold-selection, MATLAB Central File Exchange. Retrieved August 30, 2023.
2, More state of the art methods should be tested besides the gradient boosting method and the Unet method. In addition, I think the gradient boosting also belongs to the neural network method.
3, English should be improved.
Comments on the Quality of English LanguageShould be improved significantly.
Author Response
Dear Reviewer,
thank you for your valuable comments. We carefully studied each of them and tried to take them into account in the manuscript. Changes (except minor changes and grammar) are highlighted in green.
1, More image segmentation methods should be tested and compared with the used one. For instance, the following methods:
https://www.mathworks.com/matlabcentral/fileexchange/56371-zhenzhou-threshold-selection, MATLAB Central File Exchange. Retrieved August 30, 2023.
2, More state of the art methods should be tested besides the gradient boosting method and the Unet method. In addition, I think the gradient boosting also belongs to the neural network method.
In section 4.4., we have added comparison with a widely used semantic segmentation method, based on threshold segmentation. Other methods require significant refinement to be compared with ours, since they do not imply classification based on the thickness of the bitumen layer. This could be a topic for an independent study.
3, English should be improved.
Done.
Thank you for your attention and collaboration! We hope the changes made have significantly improved the structure and readability of the manuscript. We welcome your comments on the updated version.
Вest regards,
the authors.
Reviewer 3 Report
Comments and Suggestions for Authors
a) In the abstract, the authors wrote a lot of redundant information and do not objectively present the methodology they used. Just as they were objective when briefly presented the results, they should do the same for the main steps of the methodology described in the abstract.
b) In chapter 1, the authors wrote multiple statements without substantiating the sentences. It is necessary to include references that provide justification for the several statements.
c) Figures that have already been published in the literature and were not created by the authors (example: figure 2) must be removed to avoid copyright infringement.
d) The state of the art is disproportionate, the first works ([1] and [2]) explain the methodology in detail, but in the following works (Example [7], [8], [9], [11] only approximately 4 lines for each work. The works presented do not include the quantitative result achieved in each work.
e) The state of the art regarding references [13] [14] and [15] are too summarized. What methodology is used in each work? What differences exist in these three methodologies? What quantitative results were obtained?
f) The methodology presented is composed of hardware and software, but the authors highlight the software part as being more relevant. There should be a flow diagram for the software, where each step is accompanied by a brief explanation of each step.
g) Although it is positive that a methodology is accompanied with images to better understand the methodology, the authors (in the methodology chapter) apparently make a mixture of methods and results, as figures 11, 12 and 13 are images of practical cases and as such should be in the results.
h) The construction of table 1 should be better explained, namely how the values presented in this table appear; the authors mention that they are quartile values, but how do the values appear that, later, in a statistical analysis correspond to the four quartiles? Being specific values of materials, it is questionable whether this information should be in methods or results.
i) The text in section 3.4.2 becomes repetitive because several words are being repeated, and also because the number of bullets takes up almost a page. Authors can make text more appealing, for example, instead of “● The average RGB value of the minimum image around the (15, 15) pixel; ● The average HSV value of the minimum image around the (15, 15) pixel; ● The median RGB value of the minimum image around the (15, 15) pixel; ● The median HSV value of the minimum image around the (15, 15) pixel;” can write “minimum image around the (15, 15) pixel: extracted the average RGB value, the average HSV value , the median RGB value, the median HSV value”.
j) Again in section 3.4.3 the authors describe the UNet architecture with numerical information that corresponds to results and not to the methodology (example: size of the input image, number of epochs, etc.).
k) Table 2 presents the training and validation results. This table should be in the results chapter and not in methodology. The same occurs for tables 3 and 4.
l) Chapter 4 Results takes up half a page! Unacceptable in a scientific work that is intended to be published in a highly prestigious international journal, MDPI Sensors.
m) The conclusion chapter is too small. It is not enough to state that a patent has been registered for the work to be published. It is necessary to re-write the conclusions, including the following: brief framing of the problem, summary explanation of what was done at the hardware level, summary explanation (1 paragraph) for each approach implemented at the software level. And also make a summary description of the quantitative results, present the conclusions, highlight what you consider to be innovation (what this work presents as being new).
n) Analyzing the references, “APP” in [8] should be written in full text, also it is seen that a substantial number of references (almost half of the references) of works were published in conferences. On the other hand, about half of the references are more than 5 years old, which suggests that this work is based on old papers.
Author Response
Dear Reviewer,
thank you for your valuable comments. We carefully studied each of them and tried to take them into account in the manuscript. Changes (except minor changes and grammar) are highlighted in green.
a) In the abstract, the authors wrote a lot of redundant information and do not objectively present the methodology they used. Just as they were objective when briefly presented the results, they should do the same for the main steps of the methodology described in the abstract.
We have rewritten the abstract to make it more sufficient and consistent.
b) In chapter 1, the authors wrote multiple statements without substantiating the sentences. It is necessary to include references that provide justification for the several statements.
We revised the text of Introduction and added references to related works.
c) Figures that have already been published in the literature and were not created by the authors (example: figure 2) must be removed to avoid copyright infringement.
Done.
d) The state of the art is disproportionate, the first works ([1] and [2]) explain the methodology in detail, but in the following works (Example [7], [8], [9], [11] only approximately 4 lines for each work. The works presented do not include the quantitative result achieved in each work.
We have expanded and deepened the review of related works and added links to more recent surveys on the topic.
e) The state of the art regarding references [13] [14] and [15] are too summarized. What methodology is used in each work? What differences exist in these three methodologies? What quantitative results were obtained?
We have expanded this part of the review.
f) The methodology presented is composed of hardware and software, but the authors highlight the software part as being more relevant. There should be a flow diagram for the software, where each step is accompanied by a brief explanation of each step.
In section 3.2, we added flow diagram for software operation and explanation on how the device works.
g) Although it is positive that a methodology is accompanied with images to better understand the methodology, the authors (in the methodology chapter) apparently make a mixture of methods and results, as figures 11, 12 and 13 are images of practical cases and as such should be in the results.
We have reworked the structure of the manuscript and placed all the parts regarding experiments in section 4.
h) The construction of table 1 should be better explained, namely how the values presented in this table appear; the authors mention that they are quartile values, but how do the values appear that, later, in a statistical analysis correspond to the four quartiles? Being specific values of materials, it is questionable whether this information should be in methods or results.
Done.
i) The text in section 3.4.2 becomes repetitive because several words are being repeated, and also because the number of bullets takes up almost a page. Authors can make text more appealing, for example, instead of “● The average RGB value of the minimum image around the (15, 15) pixel; ● The average HSV value of the minimum image around the (15, 15) pixel; ● The median RGB value of the minimum image around the (15, 15) pixel; ● The median HSV value of the minimum image around the (15, 15) pixel;” can write “minimum image around the (15, 15) pixel: extracted the average RGB value, the average HSV value , the median RGB value, the median HSV value”.
For better readability, we have placed the description of the features in Table 2.
j) Again in section 3.4.3 the authors describe the UNet architecture with numerical information that corresponds to results and not to the methodology (example: size of the input image, number of epochs, etc.).
Done (see answer to comment g).
k) Table 2 presents the training and validation results. This table should be in the results chapter and not in methodology. The same occurs for tables 3 and 4.
Done (see answer to comment g).
l) Chapter 4 Results takes up half a page! Unacceptable in a scientific work that is intended to be published in a highly prestigious international journal, MDPI Sensors.
You are absolutely right, thank you for highlighting this significant shortcoming. We extended the experiment result section.
m) The conclusion chapter is too small. It is not enough to state that a patent has been registered for the work to be published. It is necessary to re-write the conclusions, including the following: brief framing of the problem, summary explanation of what was done at the hardware level, summary explanation (1 paragraph) for each approach implemented at the software level. And also make a summary description of the quantitative results, present the conclusions, highlight what you consider to be innovation (what this work presents as being new).
We have expanded the Conclusion according to your suggestions.
n) Analyzing the references, “APP” in [8] should be written in full text, also it is seen that a substantial number of references (almost half of the references) of works were published in conferences. On the other hand, about half of the references are more than 5 years old, which suggests that this work is based on old papers.
The references were checked and extended with more recent surveys.
Thank you for your attention and collaboration! We hope the changes made have significantly improved the structure and readability of the manuscript. We welcome your comments on the updated version.
Вest regards,
the authors.
Round 2
Reviewer 2 Report
Comments and Suggestions for Authors
The revision did not make significant improvement.
Comments on the Quality of English LanguageEnglish should be improved.
Author Response
Dear Reviewer,
We have studied and described the method you suggested in the Related Works Section. Unfortunately, we cannot reproduce it exactly in our experiments due to the peculiarities of hardware, and some other circumstances. However, for comparison, we implemented another threshold segmentation approach (TS) and presented experimental data in sections 4.4, 4.5.
There are several disadvantages of the algorithms of this type in the described problem. Firstly, the algorithm considers exclusively the intensity space of the image and ignores the color division of classes in the image, which is a significant factor in our case. Secondly, the most significant drawback of this approach is the need to provide the algorithm with information about the number of classes for segmentation in the image. This problem does not arise when solving the problem of simple binary segmentation. In our case of multi-class segmentation, certain classes may not be present in the image. In such cases, the algorithm must be able to independently determine only those classes that are present in the image.
Given the applied nature of our research, we did not strive to compare all known approaches to the image segmentation problem. Based on the analysis of modern reviews, we have chosen the Fully Convolutional Networks (FCN) approach as the most appropriate for the described problem. One of the most popular, relatively simple, and effective options for the FCN architecture is the U-Net architecture, which was chosen to implement semantic segmentation. In order to extend the study, XGBoost was chosen as an alternative approach with sufficient generalization capability. In addition, we partially expanded the literature review.
Regarding the English language improving, we asked a professional translator to proofread our manuscript.
We hope that the changes made will be sufficient.
Best regards,
the authors
Reviewer 3 Report
Comments and Suggestions for Authors
The authors have made major improvements to the scientific quality of the document as well as the presentation quality.
I am satisfied with the work of the authors.
Author Response
Dear Reviewer,
thank you for your answer!
Best regards,
the authors